# bZIP Transcription Factor PavbZIP6 Regulates Anthocyanin Accumulation by Increasing Abscisic Acid in Sweet Cherry

**DOI:** 10.3390/ijms251810207

**Published:** 2024-09-23

**Authors:** Shilin Gai, Bingyang Du, Yuqin Xiao, Xiang Zhang, Maihemuti Turupu, Qisheng Yao, Xinyu Wang, Yongzhen Yan, Tianhong Li

**Affiliations:** Frontiers Science Center for Molecular Design Breeding, College of Horticulture, China Agricultural University, Beijing 100193, China; gshilincau@outlook.com (S.G.); tzdubingyang@163.com (B.D.); s20193172433@cau.edu.cn (Y.X.); 18306391375@163.com (X.Z.); mehmud01@163.com (M.T.); s20223172961@cau.edu.cn (Q.Y.); 18562580137@163.com (X.W.); yanyongzhen02@163.com (Y.Y.)

**Keywords:** sweet cherry, *PavbZIP6*, anthocyanin, abscisic acid

## Abstract

Basic leucine zipper (bZIP) transcription factors (TFs) play a crucial role in anthocyanin accumulation in plants. In addition to bZIP TFs, abscisic acid (ABA) increases anthocyanin biosynthesis. Therefore, this study aimed to investigate whether bZIP TFs are involved in ABA-induced anthocyanin accumulation in sweet cherry and elucidate the underlying molecular mechanisms. Specifically, the BLAST method was used to identify *bZIP* genes in sweet cherry. Additionally, we examined the expression of ABA- and anthocyanin-related genes in sweet cherry following the overexpression or knockdown of a *bZIP* candidate gene. In total, we identified 54 *bZIP*-encoding genes in the sweet cherry genome. *Basic leucine zipper 6* (*bZIP6*) showed significantly increased expression, along with increased anthocyanin accumulation in sweet cherry. Additionally, yeast one-hybrid and dual-luciferase assays indicated that PavbZIP6 enhanced the expression of anthocyanin biosynthetic genes (*PavDFR*, *PavANS*, and *PavUFGT*), thereby increasing anthocyanin accumulation. Moreover, PavbZIP6 interacted directly with the *PavBBX6* promoter, thereby regulating *PavNCED1* to promote abscisic acid (ABA) synthesis and enhance anthocyanin accumulation in sweet cherry fruit. Conclusively, this study reveals a novel mechanism by which *PavbZIP6* mediates anthocyanin biosynthesis in response to ABA and contributes to our understanding of the mechanism of *bZIP* genes in the regulation of anthocyanin biosynthesis in sweet cherry.

## 1. Introduction

Sweet cherry (*Vitis vinifera* L.) is an economically valuable fruit with increased consumer demand. Notably, skin colour, sweetness, firmness, and acidity are key quality attributes of sweet cherries. Particularly, skin colour, ranging from deep red to light yellow, is crucial, as it influences the fruit’s economic value and consumer purchasing decisions [1]. Therefore, cherry colour plays a critical role in determining the commercial viability of grape products [2]. 

Anthocyanins play a crucial role in determining the colour of sweet cherries. Importantly, anthocyanins are essential for plant function and have beneficial effects on human health, including antioxidative, antimutagenic, and anticancer effects [3,4,5]. Therefore, enhancing anthocyanin accumulation in plants is a major research focus. Anthocyanins are important secondary metabolites belonging to the flavonoid class and are synthesised through the glycosylation of anthocyanidins. Additionally, anthocyanins are widely distributed in plant roots, stems, flowers, and fruits. Notably, the biosynthetic pathway of anthocyanins involves over 20 enzymatic steps that are catalysed by a series of enzymes, which are regulated by structural and regulatory genes, including chalcone synthase (*CHS*), flavanone 3-hydroxylase (*F3H*), flavonoid 3′-hydroxylase (*F3′H*), dihydroflavonol 4-reductase (*DFR*), anthocyanidin synthase (*ANS*), and flavonoid 3-O-glucosyltransferase (*UFGT*) [6]. Sweet cherries are characterized by their anthocyanin profile, which prominently features two major compounds: cyanidin-3-glucoside and cyanidin-3-rutinoside. Additionally, they contain smaller amounts of peonidin-3-rutinoside and pelargonidin-3-rutinoside as minor anthocyanins [7,8]. In many plants, these structural genes are regulated by the MBW regulatory complexes composed of R2R3-MYB transcription factors, bHLH proteins, and WD40 repeat proteins [9]. Research indicates that MYB TFs are critical in regulating anthocyanin biosynthesis [10].

In addition to the MYB TF family, research evidence indicates that the basic region-leucine zipper (bZIP) TF family plays a significant role in anthocyanin biosynthesis. The conserved bZIP domain consists of approximately 40–80 amino acid residues and comprises two parts: a highly conserved DNA-binding basic region composed of 20 amino acids and a relatively diverse leucine zipper region [11]. The basic region, which is rich in basic amino acids, is located at the C-terminus and binds to DNA sequences, specifically through a fixed N-x7-R/K structure. The leucine zipper region at the N-terminus consists of several heptad repeats or hydrophobic amino acid residues such as leucine, isoleucine, and valine. The primary function of this domain is the formation of dimers via leucine zipper structures [12,13]. In addition to the bZIP domain, bZIP TFs contain other conserved domains that serve as transcription activation factors. For example, two conserved motifs, R/KxxS/T and S/TxxD/E, have been identified as phosphorylation sites for Ca^2+^-independent protein kinases and casein kinase II, respectively [12]. Domains rich in proline, glutamine, and acidic residues have also been implicated in the transcriptional activation of *bZIP* genes [14].

*HYPOCOTYL5* (*HY5*) is a member of the *bZIP* gene family and is activated in a light-dependent manner to promote pigment accumulation. HY5 directly binds to the G-box or ACE-box motifs of *MYB* factors, including those producing anthocyanins, such as *PIGMENT1* (*PAP1*), flavonol glycosides (*MYB12* and *MYB111*), and *MYB-like* structures (*MYBD*), thereby enhancing their gene expression [15,16,17,18,19]. Anthocyanin accumulation is promoted by MdbZIP44, which enhances the interaction between *MdMYB1* and downstream gene promoters [20]. Notably, the overexpression of *PgbZIP16* and *PgbZIP34* from pomegranate promoted anthocyanin accumulation in the leaves of *Arabidopsis* [21]. Additionally, *VvbZIP36* plays a crucial role in regulating anthocyanin synthesis in grapes by ensuring a balance in the production of stilbenes, lignans, flavonols, and anthocyanins [22]. Moreover, *HlbZIP1A* and *HlbZIP2* expression increased flavonol glycosides, phenolic acids, and anthocyanin contents through the activation of the *CHS_H1* promoter in *Humulus lupulus* [23]. Although the bZIP protein family has been shown to play key roles in anthocyanin synthesis in some plant species [22,24,25], its role in the regulation of anthocyanin synthesis in sweet cherry remains unclear.

ABA plays a distinct role in promoting seed and bud dormancy, as well as in regulating stress responses in various plant species [26,27,28]. ABA is crucial for anthocyanin accumulation, which enhances colour richness during fruit ripening [29]. ABA treatment increased the transcription of *FaMYB10* and promoted anthocyanin accumulation by upregulating genes involved in anthocyanin biosynthesis in strawberries [30]. Additionally, *MdMYB1* regulated ABA-induced anthocyanin biosynthesis in apples [24]. Moreover, *PacMYBA* plays a key role in ABA-mediated anthocyanin accumulation in sweet cherries [31]. However, the regulatory pathways of ABA signalling in anthocyanin biosynthesis in sweet cherry require further elucidation.

Therefore, this study aimed to investigate whether bZIP TFs are involved in ABA-induced anthocyanin accumulation in sweet cherry and elucidate the underlying molecular mechanisms. Specifically, we identified members of the bZIP TF family in sweet cherries and analysed their conserved domains, evolutionary relationships, and cis-acting elements using bioinformatic approaches.

## 2. Results

### 2.1. Identification and Characterization of the bZIP TF Family in Sweet Cherry

In total, 54 members of the bZIP family were identified in the genome of the sweet cherry cultivar ‘Tieton’. For subsequent analyses, the sequences were renamed according to their chromosomal positions (Table 1). Additionally, we analysed the physicochemical properties of the bZIP TFs using ExPASy online tools and found that the molecular weights of the proteins range from 7872.53 to 111,776.79 Da, with theoretical isoelectric points of 3.73–11.1. Moreover, the protein lengths varied from 74 to 1000 amino acids, with the shortest being PavbZIP41 (74 aa) and the longest being PavbZIP10 (1000 aa). Overall, these findings provide a theoretical basis for further research on the purification, activity, and function of PavbZIP. Subcellular localisation predictions for each member indicated that all members of the *bZIP* gene family were expressed in the nucleus. Based on these results, we investigated the role of PavbZIP proteins in transcriptional regulation and their potential involvement in anthocyanin synthesis in sweet cherry.

### 2.2. Phylogenetic Analysis of PavbZIP Proteins in Sweet Cherry

To explore homologous evolutionary relationships and classifications within the bZIP family, we constructed a phylogenetic tree using bZIP members from sweet cherry and *Arabidopsis*. Notably, *bZIP* gene family members in sweet cherry were divided into 13 groups, labelled A, B, C, D, E, G, H, I, K, M, and S. Additionally, these groups vary greatly in size, with the K group consisting of only one member and the F and J groups having no members at all. Importantly, the largest group was Group A, which included 13 members (Figure 1).

### 2.3. PavbZIP6 Is Closely Associated with Anthocyanin Synthesis in Sweet Cherry

To investigate the role of the bZIP TFs in anthocyanin synthesis, we examined *bZIP* mRNA expression and anthocyanin content in sweet cherry at three developmental stages: the big green (BG), yellow white (YW), and full red (FR) stages (Figure 2A). It’s important to note that the anthocyanin content in sweet cherries rises significantly as the fruit matures, with the content at the FR stage being 26.2 times higher than that at the BG stage (Figure 2B). Additionally, an expression analysis of the 54 identified *bZIP* genes across the three stages showed significant upregulation of only *bZIP6*, with its expression at the FR stage being 4.5 times higher than that at the BG stage (Figure 2C). Collectively, these findings suggest that bZIP6 plays a crucial role in regulating anthocyanin synthesis in sweet cherry.

### 2.4. Functional Characterization of PavbZIP6

To further investigate the function of PavbZIP6, we transiently expressed the fusion protein of PavbZIP6 and green fluorescent protein (GFP) in tobacco epidermal cells and observed the GFP’s fluorescence signals 48 h later. Notably, the fluorescence signal of the PavbZIP6 fusion protein in sweet cherry was localised to the nucleus, indicating its role as a nuclear-localised transcription factor (Figure 3A). Therefore, we validated PavbZIP6 transcriptional self-activation in a yeast system. Fusion with the LexA DNA-binding domain activated the transcription of the *LacZ* reporter gene in the presence of PavbZIP6, confirming that PavbZIP6 is a transcriptional activator (Figure 3B).

### 2.5. PavbZIP6 Promotes Anthocyanin Accumulation in Sweet Cherry Fruits

To further investigate the function of PavbZIP6, an Agrobacterium-mediated transient injection system was employed for the transient overexpression and silencing of *PavbZIP6* in sweet cherry ‘Rainier’ fruits (Figure 4A). Real-time PCR revealed a significant increase in *PavbZIP6* expression in fruits overexpressing the gene compared with that in the control group, whereas *PavbZIP6* silencing showed the opposite effect (Figure 4B). Phenotypic observations indicated that *PavbZIP6* overexpression caused a darker fruit colouration relative to that of the controls, whereas silencing resulted in a lighter colouration. Consistent results were obtained from the anthocyanin content measurements (Figure 4C).

Considering that the transient overexpression of *PavbZIP6* enhanced anthocyanin accumulation in fruits, an RT-qPCR analysis was conducted to assess changes in the expression patterns of anthocyanin biosynthetic genes in transgenic sweet cherry fruits. *PavbZIP6* overexpression significantly upregulated anthocyanin biosynthesis-related genes, including *PavCHS*, *PavF3H*, *PavF3′H*, *PavDFR*, *PavANS*, and *PavUFGT* (Figure 4D), whereas *PavbZIP6* knockdown significantly downregulated the genes in sweet cherry, consistent with the observed phenotypes (Figure 4E). Collectively, these findings indicate that *PavbZIP6* overexpression promotes anthocyanin accumulation in sweet cherry fruits by modulating the expression of anthocyanin biosynthetic genes.

### 2.6. PavbZIP6 Regulates the Expression of Structural Genes in Anthocyanin Biosynthesis

To further investigate how PavbZIP6 regulates the accumulation of anthocyanins in sweet cherry, we examined significantly altered genes in *PavbZIP6*-overexpressing fruit, including *PavCHS*, *PavF3H*, *PavF3′H*, *PavDFR*, *PavANS*, and *PavUFGT*. Specifically, we cloned their full-length promoters into the pLacZi2μ vector as reporters, while the *PavbZIP6* sequence was cloned into the pB42AD vector as effectors. A yeast one-hybrid analysis revealed that co-transforming empty effectors with *PavDFR*, *PavANS*, and *PavUFGT* reporters did not result in a blue colour in the selection plates. In contrast, the co-transformation of effectors containing *PavbZIP6* with *PavDFR*, *PavANS*, and *PavUFGT* reporters consistently yielded blue colonies (Figure 5A). Overall, these results demonstrate that PavbZIP6 can directly bind to the promoters of the anthocyanin biosynthesis genes *PavDFR*, *PavANS*, and *PavUFGT*.

To further analyse the regulatory effects of *PavbZIP6* on the promoters of *PavDFR*, *PavANS*, and *PavUFGT*, we examined LUC activity in tobacco using a dual-luciferase reporter assay. Live fluorescent imaging showed weak or negligible fluorescence signals in tobacco cells co-injected with empty effectors and *PavDFR*, *PavANS*, and *PavUFGT* promoters. In contrast, strong fluorescence signals were observed in cells co-injected with effectors containing PavbZIP6 with these promoters. Notably, results consistent with the fluorescence imaging were obtained in the LUC enzyme activity assays (Figure 5B,C). Collectively, these findings indicate that PavbZIP6 enhances the expression of the anthocyanin biosynthesis genes *PavDFR*, *PavANS*, and *PavUFGT*.

### 2.7. PavbZIP6 Regulates Anthocyanin Synthesis by Controlling ABA Levels through the Modulation of PavBBX9 Expression

ABA plays a significant role in anthocyanin biosynthesis in sweet cherry fruits. For example, *PavBBX6* and *PavBBX9* promote ABA accumulation in sweet cherry fruits, thereby enhancing anthocyanin accumulation [32]. To investigate whether *PavbZIP6* regulates anthocyanin biosynthesis via ABA, we examined the mRNA expression of *PavBBX6*, *PavBBX9*, and *HY5*. The RT-qPCR revealed that *PavbZIP6* overexpression significantly upregulated the mRNA expression of *PavbZIP6*, *PavBBX6*, *PavBBX9*, and *HY5*. In contrast, *PavbZIP6* silencing significantly downregulated *PavBBX6*, *PavBBX9*, and *HY5* mRNA levels (Figure 6A). Overall, these findings suggest that PavbZIP6 promotes anthocyanin accumulation in sweet cherry fruits through the expression of these genes.

Furthermore, we explored the regulatory effects of *PavbZIP6* on *PavBBX6*, *PavBBX9*, and *HY5*. Yeast one-hybrid assays showed that the co-transformation of the effector construct containing PavbZIP6 with the reporter constructs *PavBBX9* and *HY5* did not result in blue colouration on the selection plates. However, the co-transformation of the effector construct containing *PavbZIP6* with the reporter construct *PavBBX6* alone resulted in blue colouration (Figure 6B). Moreover, the dual-luciferase reporter assay and LUC enzyme activity analysis confirmed that *PavbZIP6* positively regulated *PavBBX6* (Figure 6C,D), indicating that *PavbZIP6* may modulate anthocyanin biosynthesis by regulating *PavBBX6* expression. Additionally, we measured the ABA content in fruits following *PavbZIP6* overexpression or knockdown. *PavbZIP6* overexpression significantly increased ABA content, whereas *PavbZIP6* knockdown markedly decreased ABA content in fruits (Figure 6E). Moreover, the RT-PCR showed that *PavbZIP6* overexpression significantly upregulated ABA biosynthesis-related genes (*PavNCED1* and *PavNCED3*) in sweet cherry fruits compared with those in the control group. In contrast, *PavbZIP6* knockdown significantly downregulated *PavNCED1* expression. Collectively, these findings suggest that *PavbZIP6* regulates ABA levels and influences anthocyanin biosynthesis by modulating the expression of *PavBBX6* and *PavNCED1* genes.

## 3. Discussion

In this study, we identified 54 *bZIP* genes in the sweet cherry genome database (Table 1). Notably, the number of *bZIP* genes varies across plant species, with Arabidopsis having 78 [33], humans having 53 members [34], rice having 92 [35], and maize having 125 [36]. Differences in the number of *bZIP* genes may be attributed to variations in genome size and complexity between these species. Based on multiple clustering analyses of *Arabidopsis* bZIP proteins, PavbZIP proteins were classified into 13 subgroups (Figure 1). Group S represented the largest cluster of PavbZIP proteins in sweet cherry, similar to findings for *Arabidopsis* [33]. In *Arabidopsis*, the S subgroup is the largest bZIP cluster (17 members), typically comprising intronless genes encoding small TF proteins of approximately 20 kDa. All the members of the S subgroup preferentially form heterodimers, collectively known as the C/S1-bZIP network [33]. Whether bZIP proteins in the S subgroup of sweet cherries also function via heterodimerisation remains to be explored.

Currently, bZIP TFs play crucial roles in plant growth, development, and responses to abiotic stresses, such as seed maturation, flower development, and stress responses [37,38,39]. Importantly, the bZIP gene family has been extensively studied in plants, including *AtbZIP11/18* in *Arabidopsis* [40,41], *GsbZIP67* in soybean [42], *CabZIP25* in pepper [39], *TabZIP15* in wheat [43], and *MdbZIP44* in apples [24]. Despite the completion of whole-genome sequencing of sweet cherry, little is known about the *bZIP* gene family, particularly in relation to anthocyanin synthesis. Comprehensive studies have been undertaken on the functions of the *bZIP* genes *HY5* and *HYH* in *Arabidopsis*, tomatoes, and apples. For example, *HY5* and *HYH* act as downstream photoreceptors in the light signalling pathway in *Arabidopsis* by directly activating the expression of early biosynthetic genes (EBGs) and late biosynthetic genes (LBGs) and positively regulating the transcriptional activation of *AtPAP1* [38,44,45]. In apples, the bZIP TF gene *MdHY5* directly promoted the expression of *MdMYB10* and *MdMYB1* and positively regulated anthocyanin accumulation by enhancing interactions with downstream target genes [24,42,46]. *SlHY5* knockdown in tomatoes downregulated anthocyanin accumulation [47]. In tomatoes, the bZIP transcription factor SlAREB1 regulates anthocyanin biosynthesis in seedlings under low temperatures through an ABA-dependent pathway [48]. The MpbZIP9 transcription factor regulates the synthesis of anthocyanins in fruit skin and acts as a positive regulator by promoting anthocyanin biosynthesis through the activation of *MpF3′H* expression [49]. The ABA-induced NAC transcription factor MdNAC1 interacts with bZIP-type transcription factors to promote anthocyanin synthesis in red-fleshed apples [50]. In the present study, we examined the expression of the 54 identified bZIP TFs at different developmental stages of sweet cherries. Although anthocyanin accumulation increased in the skin of the ‘Tieton’ cultivar along with *PavbZIP6* upregulation, *HY5* expression was not enhanced (Figure 2). A further analysis of *PavbZIP6* revealed its nuclear localisation and transcriptional self-activation (Figure 3), suggesting that PavbZIP6 may function as a transcriptional regulator of other genes in the nucleus. The transient overexpression of *PavbZIP6* in ‘Rainier’ cherry caused a darker fruit colour compared to that of the controls, whereas *PavbZIP6* silencing resulted in fruits with a lighter colour (Figure 4). Collectively, these results indicate that *PavbZIP6* overexpression enhances anthocyanin accumulation in fruits. Yeast one-hybrid and dual-luciferase assays further demonstrated that *PavbZIP6* enhanced the expression of the anthocyanin biosynthesis genes *PavDFR*, *PavANS*, and *PavUFGT*, thereby increasing anthocyanin accumulation.

Recent studies have demonstrated that bZIP TFs play crucial and diverse roles in the ABA-mediated regulation of plant growth, development, and abiotic stress responses [48]. ABA-induced anthocyanin accumulation has also been reported in many species, such as apple [20], Arabidopsis [51], wolfberry [52], and maize [53]. In *Arabidopsis*, *bZIP16* directly regulates ABA- and light-responsive genes to promote seed germination and hypocotyl elongation [54]. In apples, *MdbZIP44* enhances anthocyanin biosynthesis in an ABA-dependent manner [24]. *MdNAC1* forms a complex with *MdbZIP23* in the ABA signalling pathway, promoting the expression of *MdMYB10* and *MdUFGT* and the accumulation of anthocyanins [50]. As a key regulatory factor in signal transduction, ABA is closely associated with anthocyanin biosynthesis in various species [28,55,56,57,58]. However, the molecular mechanisms by which the bZIP TFs influence ABA-induced anthocyanin synthesis in sweet cherry remain unclear. In the present study, *PavbZIP6* overexpression in cherry fruit significantly upregulated *PavBBX6*, *PavBBX9*, and *HY5*, whereas *PavbZIP6* knockdown significantly downregulated these genes (Figure 6A). Yeast one-hybrid and dual-luciferase reporter assays confirmed that PavbZIP6 positively regulates *PavBBX6* (Figure 6B–D), indicating that *PavbZIP6* regulates anthocyanin biosynthesis by modulating *PavBBX6* expression. A previous study showed that *PavBBX6* promoted ABA accumulation in sweet cherry fruits by regulating the ABA biosynthesis gene *PavNCED1*, thereby enhancing anthocyanin accumulation [29]. Notably, our measurements of ABA content and *PavNCED1*, *PavNCED2*, and *PavNCED3* expression in fruits following *PavbZIP6* overexpression or silencing confirmed these findings (Figure 6E,F).

This study identified all bZIP genes in sweet cherry fruits and identified *PavbZIP6* as a TF closely associated with anthocyanin synthesis. Additionally, we demonstrated that the protein encoded by *PavbZip6* binds directly to the promoters of the anthocyanin biosynthesis genes *PavDFR*, *PavANS,* and *PavUFGT*, thereby promoting their expression and regulating anthocyanin biosynthesis (Figure 5). Moreover, *PavbZIP6* can bind directly to the promoter of *PavBBX6* to promote ABA synthesis (Figure 6). Conclusively, this study confirmed that PavbZIP6 is a positive regulatory factor that mediates anthocyanin synthesis in sweet cherry, laying the groundwork for further explorations of the molecular mechanisms underlying anthocyanin biosynthesis in sweet cherry.

## 4. Materials and Methods

### 4.1. Plant Materials and Growth Conditions

The ‘Tieton’ sweet cherry (*Prunus avium* L.) was cultivated at the Beijing Forestry Fruit Tree Research Institute. Fruit samples were collected at three developmental stages: BG (approximately 15 days after flowering), YW (approximately 22 days after flowering), and FR (approximately 28 days after flowering). The collected fruits were visually similar in appearance, with uniform size and without bruising. Thirty fruits were selected for anthocyanin analysis, while the remaining fruits were frozen in liquid nitrogen and subsequently stored at −80 °C for future analysis. Each developmental stage was replicated three times, with each replicate consisting of 10 cherry fruits.

### 4.2. Mining of PavbZIP Genes in the Prunus avium Genome and Phylogenetic Tree Analysis

To identify the bZIP transcription factor family members in cherry (*Prunus avium*), we used the BLAST method to compare the amino acid sequences of all bZIP genes from Arabidopsis thaliana with the cherry genome (http://cherry.kazusa.or.jp/, accessed on 16 October 2023). Candidate genes were identified and screened using the CDD database (https://www.ncbi.nlm.nih.gov/Structure/cdd/wrpsb.cgi, accessed on 3 October 2023) and the Pfam database (http://pfam.xfam.org/search accessed on 12 November 2023). Molecular weight and isoelectric point (pI) calculations were performed using ExPASy (https://web.expasy.org/protparam/, accessed on 25 November 2023).

Additionally, we constructed a phylogenetic tree using the neighbour-joining method in MEGA 6.0 software to compare PavbZIP proteins with bZIP proteins from Arabidopsis, with bootstrap values set to 1000 replications.

### 4.3. Determination of Anthocyanin Content

After grinding sweet cherry fruit in liquid nitrogen, 1.0 g of the sample was soaked in 5 mL of methanol:hydrochloric acid (999:1, *v*/*v*) solution. Anthocyanins were extracted from the sample overnight at −20 °C in darkness, followed by centrifugation to obtain the supernatant. Finally, anthocyanin content was determined using the pH differential method. The absorbance of each sample was measured at 510 and 710 nm using a UV-visible spectrophotometer (Shimadzu, Kyoto, Japan) in buffers at pH of 1.0 and pH of 4.5. The anthocyanin content was calculated as follows: A = {[(A510–A710) pH1.0] − [(A510–A710) pH4.5]}.

### 4.4. RT-qPCR 

Total RNA was extracted using a plant total RNA extraction kit (Huayueyang Biotechnology, Beijing, China). cDNA synthesis was performed with a reverse transcription kit (TaKaRa Biotechnology, Beijing, China). All primer sequences used in the experiments are listed in Appendix A. qRT-PCR was conducted using SYBR Premix Ex Taq (Kangwei Century Biotechnology, Beijing, China). Each experiment was performed in triplicate. Relative gene expression was calculated using the 2^−ΔΔCt^ method, with *PavActin* as the reference gene.

### 4.5. Subcellular Localization

The full-length open reading frame of PavbZIP6 was amplified from sweet cherry fruit and cloned into the plant expression vector pCAMBIA1302, which contains a green fluorescent protein (GFP) tag under the control of the 35S promoter of cauliflower mosaic virus CaMV35S promoter. Green fluorescence was observed using a laser scanning confocal microscope (FV31-HSD, OLYMPUS, Tokyo, Japan).

### 4.6. Self-Activation Activity Analysis

Briefly, EGY48 yeast competent cells were mixed with 3 μg of carrier DNA, 300 ng of LexA vector plasmid, and 300 ng of H18 LacZ-2μ reporter plasmid. After adding 300 mL of PEG lithium acetate solution (PEG/LiAc), the mixture was vortexed and incubated in a 45 °C water bath for 30 min, with intermittent vortexing every 10 min. Thereafter, the cells were plated on SD-Trp-Ura medium and cultured at 30 °C for 3 days. Colonies were selected based on size and streaked on SD-Trp-Ura medium containing X-Gal at 30 °C for 1–2 days to observe colour development.

### 4.7. Field Transient Infiltration of Sweet Cherry Fruit

Briefly, the promoter of *PavbZIP6* was amplified and cloned into the pCAMBIA1301 and pFGC5941 vectors to generate the overexpression vector, *PavbZIP6*-pCAMBIA1301, and the RNAi interference vector, *PavbZIP6*-pFGC5941. Recombinant vectors were transformed into *Agrobacterium* tumefaciens, which were then used to infiltrate sweet cherry ‘Rainier’ fruits at the large green fruit stage (approximately 20 days after flowering). Specifically, the respective bacterial solutions and corresponding empty vector solutions were slowly injected into the fruit near the pedicel to ensure thorough infiltration of the fruit using a microinjector. After injection, the fruits were bagged and labelled, and the bags were removed every other day. Samples were collected at the ripening stage of sweet cherry ‘Rainier’ fruits for analysis.

### 4.8. Yeast One-Hybrid Assay

Briefly, the coding region of *PavbZIP6* was cloned into the pB42AD vector. Additionally, the promoter regions of the genes involved in the experiment were cloned into the pLacZ-2μ vector. Thereafter, yeast cells were transformed with these constructs, and larger colonies were selected for identification. Finally, colonies on SD-Trp-Ura medium containing X-Gal were incubated at 30 °C for 1–2 days to observe colour development.

### 4.9. Transient Expression Assay in Tobacco Leaves

The promoter regions of the relevant genes were amplified and cloned into the pGreen II 0800-LUC vector. The full-length open reading frame (ORF) of PavbZIP6 was inserted into the effector vector pGreen II 0029 62-SK to create a recombinant vector. This recombinant vector was then transformed into Agrobacterium tumefaciens strain GV3101 (Psoup) and used to infiltrate tobacco leaves. After 2 days of infiltration, luciferin was uniformly sprayed onto the leaves for luminescence detection (Tanon-5200Multi, Tanon, Shanghai, China). Tobacco leaf samples were collected, and their fluorescence activity was assessed using a fluorescence detection kit (Promega, Beijing, China).

### 4.10. Measurement of ABA Content

Briefly, the ABA contents of the sweet cherry varieties ‘Tieton’ and ‘Rainier’ were measured using enzyme-linked immunosorbent assay (ELISA). All samples were ground in liquid nitrogen, and 0.5 g of each sample was extracted in 1× PBS buffer. The extraction followed the procedures outlined in the instructions of Abscisic Acid Assay Kit (Shanghai Jingkang, Shanghai, China), and all samples were analysed in triplicate.

### 4.11. Statistical Analysis

All statistical analyses were performed using SPSS 20.0 software. Statistical significance was determined using one-way ANOVA followed by Duncan’s multiple range test or Student’s *t*-test.

## Figures and Tables

**Figure 1 ijms-25-10207-f001:**
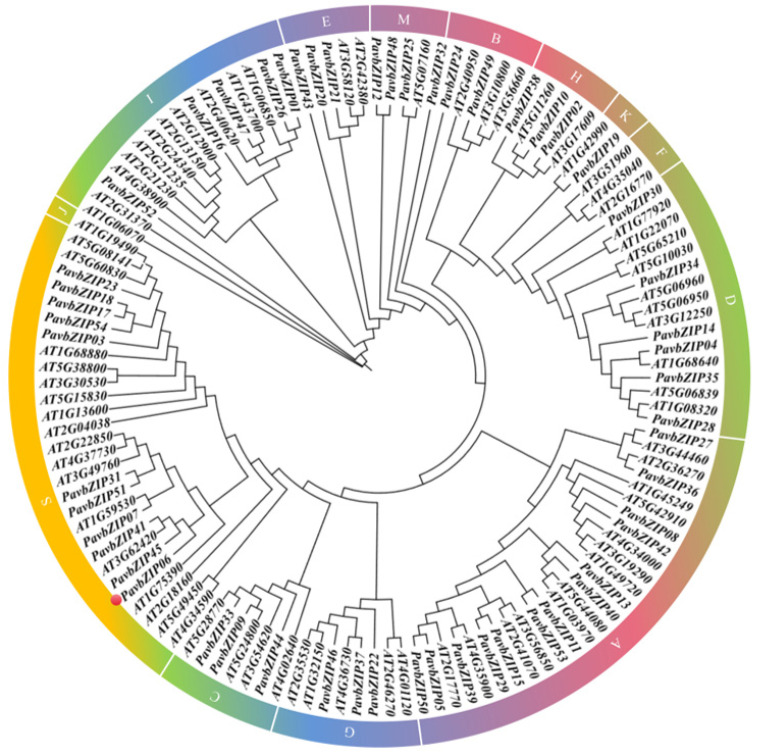
Phylogenetic analysis of PavbZIP6 protein in sweet cherry (*Prunus avium*). The protein indicated by the marked red dot is PavbZIP6.

**Figure 2 ijms-25-10207-f002:**
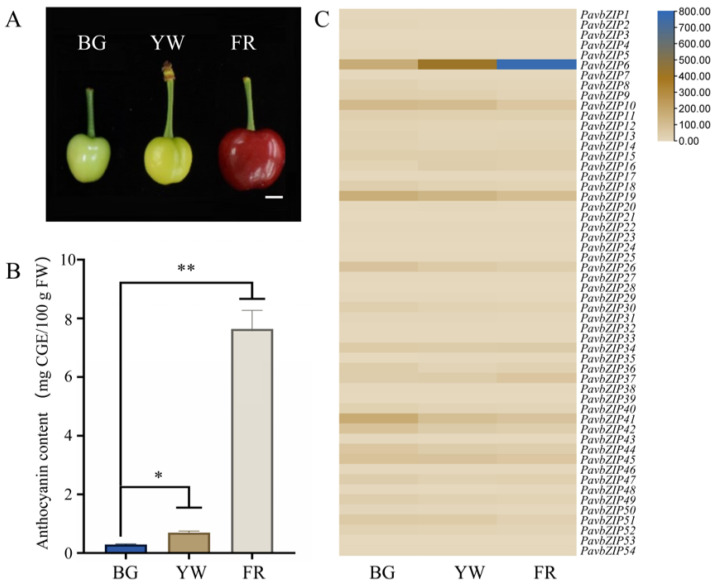
Analysis of anthocyanin content and bZIP gene family expression in sweet cherry fruits at different developmental stages. (**A**) Phenotypes of sweet cherry ‘Tieton’ fruit at different developmental stages. Scale bar: 1 cm. (**B**) Anthocyanin content of sweet cherry during three stages of fruit development: big green (BG); yellow white (YW); and full red (FR). Data are expressed as means ± standard deviation (SD) of three measurements from at least 10 sweet cherry fruits, and significant differences were assessed (* *p* < 0.05, ** *p* < 0.01). (**C**) Expression analysis of the *PavbZIP* gene family in sweet cherry ‘Tieton’ fruits at different developmental periods.

**Figure 3 ijms-25-10207-f003:**
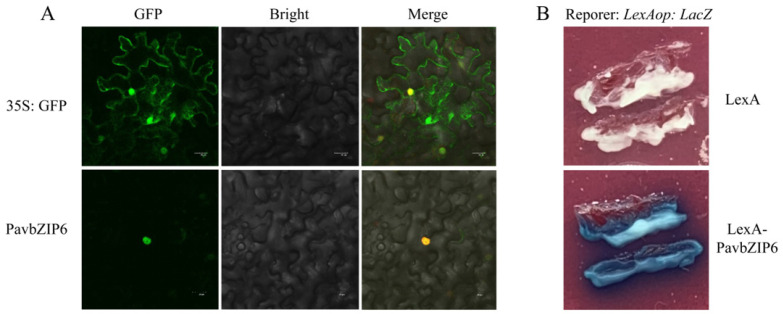
Subcellular localisation and self-activation activity analysis of PavbZIP6. (**A**) Subcellular localisation analysis of PavbZIP6 protein; Scale bar = 50 μm. Green represents GFP and the fusion protein of PavbZIP6 with GFP. Yellow indicates the color resulting from the combination of nuclear-localized *mCherry* and GFP fluorescence. (**B**) Analysis of the self-activating activity of the PavbZIP6 protein.

**Figure 4 ijms-25-10207-f004:**
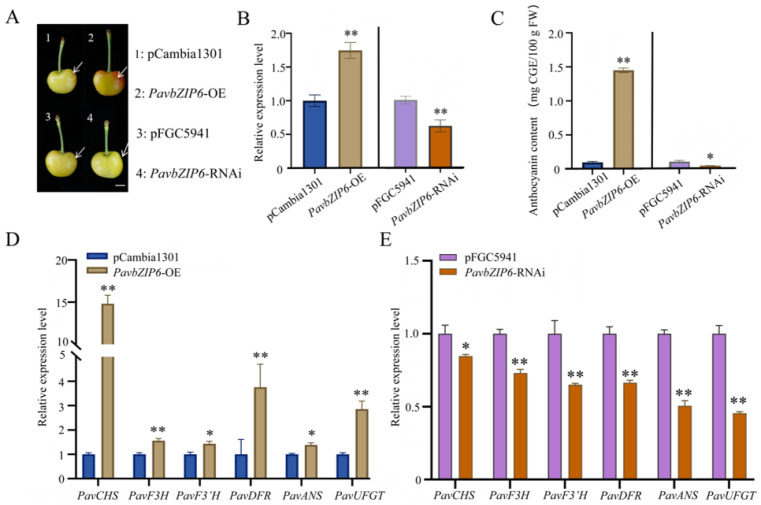
Generation of *PavbZIP6* overexpression and knockdown fruits. (**A**) Phenotypic analysis of sweet cherry following transient overexpression and silencing of *PavbZIP6*; scale bar: 1 cm. (**B**) Expression analysis of *PavbZIP6* in transgenic fruits. (**C**) Determination of anthocyanin content in sweet cherry fruit following *PavbZIP6* overexpression and silencing. (**D**) Expression analysis of genes related to anthocyanin synthesis in fruits overexpressing *PavbZIP6*. (**E**) Expression analysis of genes related to anthocyanin synthesis in fruits following *PavbZIP6* silencing. Data are expressed as means ± standard deviation (SD) of three measurements from at least 10 sweet cherry fruits, and significant differences were assessed (* *p* < 0.05, ** *p* < 0.01).

**Figure 5 ijms-25-10207-f005:**
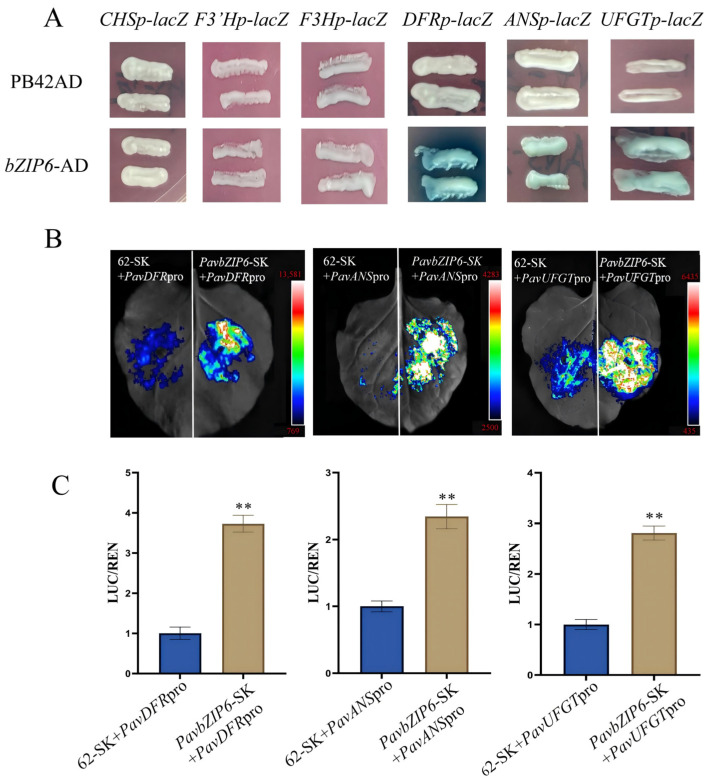
PavbZIP6 regulates the expression of *PavDFR*, *PavANS* and *PavUFGT.* (**A**) Yeast one-hybrid assay to verify that PavbZIP6 binds the promoter of the anthocyanin synthesis gene. (**B**) Transient LUC imaging assays showing that PavbZIP6 activate the transcription of the report gene. Representative images of LUC activity in N. benthamiana leaves 48 h after infiltration. (**C**) Dual-luciferase assay showing promoter activity expressed as the ratio of luciferase (LUC) to *35S::*Renilla (REN). Data are expressed as means ± standard deviation (SD; *n* = 3), and significant differences were assessed (** *p* < 0.01).

**Figure 6 ijms-25-10207-f006:**
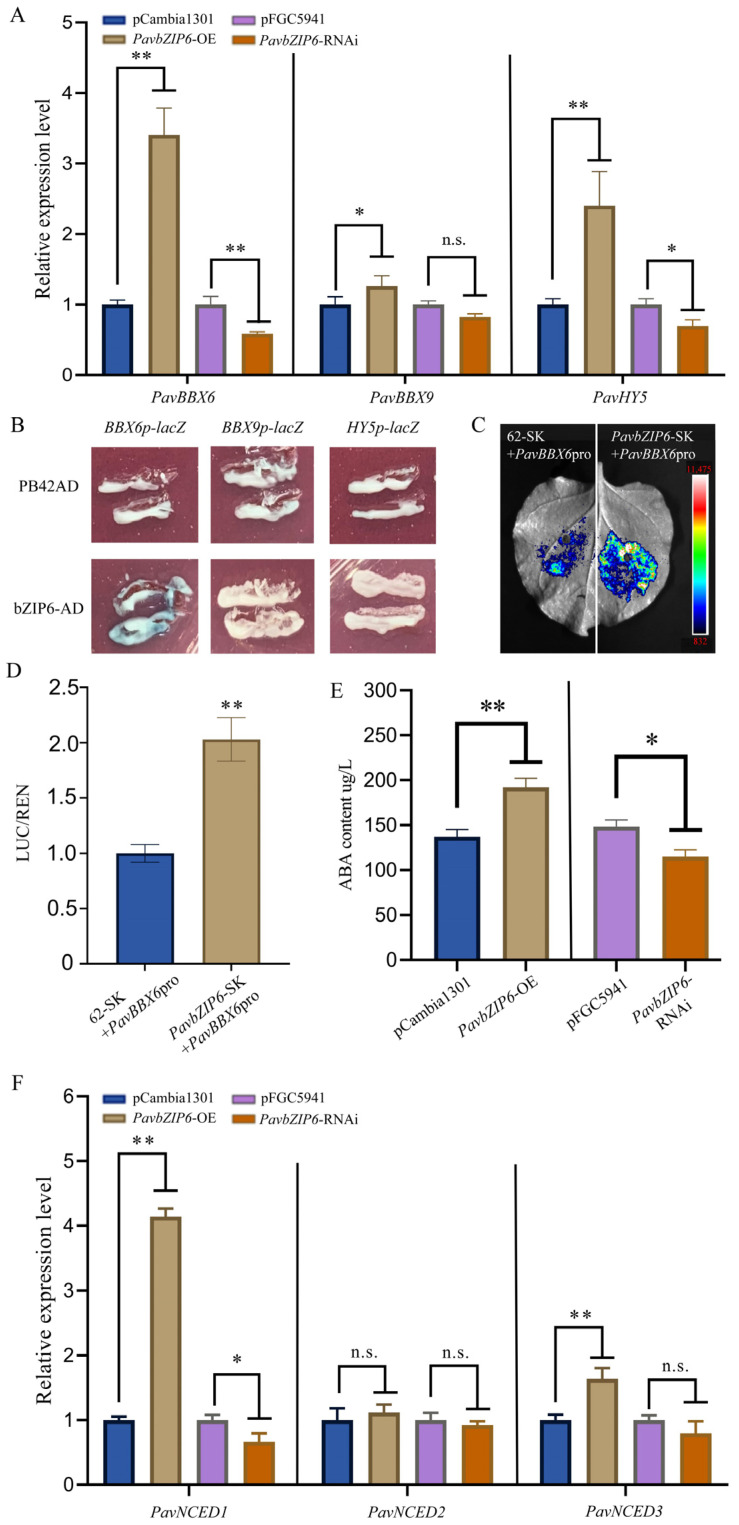
PavbZIP6 regulates the expression of PavBBX6 and the accumulation of abscisic acid (ABA). (**A**) mRNA expression of *PavBBX6/9* and *PavHY5* in transgenic fruits. (**B**) Yeast one-hybrid assay to verify the binding of *PavbZIP6* to the promoter of *PavBBX6/9* and *PavHY5.* (**C**) Transient LUC imaging assays showing that PavbZIP6 activate the transcription of the report gene. Representative images of LUC activity in *N. benthamiana* leaves 48 h after infiltration. (**D**) Dual-luciferase assay showing promoter activity expressed as the ratio of luciferase (LUC) to *35S::*Renilla (REN). (**E**) ABA contents in transgenic sweet cherry fruit and control. (**F**) mRNA expression of *PavNCED1*, *PavNCED2*, and *PavNCED3* in transgenic fruits. Data are expressed as means ± standard deviation (SD; *n* = 3), and significant differences were assessed (* *p* < 0.05, ** *p* < 0.01, n.s.: no signifcant difference).

**Table 1 ijms-25-10207-t001:** The identified *PavbZIP* genes and their corresponding information.

Gene Name	Gene ID	Group	CDS	AA	MW (Da)	pI	Subcellular Localization
*PavbZIP1*	Pav_sc0000861.1_g130.1.mk	I	1320	439	47,164.03	6.51	Nucleus
*PavbZIP2*	Pav_sc0000478.1_g270.1.mk	H	684	227	25,359.72	4.57	Nucleus
*PavbZIP3*	Pav_sc0000588.1_g350.1.br	S	615	204	23,314.85	6.51	Nucleus
*PavbZIP4*	Pav_sc0000567.1_g900.1.mk	D	1395	464	51,644.69	6.42	Nucleus
*PavbZIP5*	Pav_sc0000085.1_g210.1.mk	A	435	144	16,328.36	10.65	Nucleus
*PavbZIP6*	Pav_sc0002208.1_g520.1.mk	S	504	167	19,048.15	7.1	Nucleus
*PavbZIP7*	Pav_sc0000912.1_g090.1.mk	S	603	200	23,108.14	10.16	Nucleus
*PavbZIP8*	Pav_sc0000852.1_g810.1.mk	A	1260	419	44,868.4	10.33	Nucleus
*PavbZIP9*	Pav_sc0000800.1_g1460.1.mk	C	1044	347	37,768.46	5.42	Nucleus
*PavbZIP10*	Pav_sc0001476.1_g130.1.mk	H	3003	1000	111,776.79	7.77	Nucleus
*PavbZIP11*	Pav_sc0000030.1_g770.1.mk	A	957	318	35,694.13	6.97	Nucleus
*PavbZIP12*	Pav_sc0001196.1_g670.1.mk	M	1146	381	42,932.37	8.63	Nucleus
*PavbZIP13*	Pav_sc0001106.1_g630.1.mk	A	828	275	30,123.37	5.13	Nucleus
*PavbZIP14*	Pav_sc0001109.1_g050.1.mk	D	1191	396	43,881.26	6.81	Nucleus
*PavbZIP15*	Pav_sc0001836.1_g030.1.mk	A	969	322	35,962.58	8.17	Nucleus
*PavbZIP16*	Pav_sc0005746.1_g050.1.mk	I	1083	360	39,846.27	6.36	Nucleus
*PavbZIP17*	Pav_sc0000051.1_g030.1.br	S	609	202	23,407.84	6.68	Nucleus
*PavbZIP18*	Pav_sc0000051.1_g020.1.br	S	627	208	24,093.68	6.87	Nucleus
*PavbZIP19*	Pav_sc0000600.1_g560.1.mk	K	1017	338	37,409.07	4.42	Nucleus
*PavbZIP20*	Pav_sc0003492.1_g420.1.mk	E	1005	334	37,716.81	6.68	Nucleus
*PavbZIP21*	Pav_sc0002986.1_g270.1.mk	E	936	311	34,822.55	6.44	Nucleus
*PavbZIP22*	Pav_sc0000754.1_g090.1.mk	G	1992	663	73,814.14	8.61	Nucleus
*PavbZIP23*	Pav_sc0000358.1_g970.1.br	S	531	176	20,146.54	7.81	Nucleus
*PavbZIP24*	Pav_sc0000638.1_g900.1.mk	B	876	291	32,697.9	4.55	Nucleus
*PavbZIP25*	Pav_sc0000638.1_g910.1.mk	M	1173	390	42,986.99	9.72	Nucleus
*PavbZIP26*	Pav_sc0000206.1_g660.1.mk	I	1320	439	47,164.03	6.51	Nucleus
*PavbZIP27*	Pav_sc0001801.1_g090.1.mk	A	1152	383	42,651.34	5.12	Nucleus
*PavbZIP28*	Pav_sc0001801.1_g110.1.mk	D	1392	463	50,544.4	6.76	Nucleus
*PavbZIP29*	Pav_sc0009346.1_g010.1.mk	A	783	260	29,071.56	6.95	Nucleus
*PavbZIP30*	Pav_sc0002914.1_g170.1.mk	D	1092	363	41,013.38	6.89	Nucleus
*PavbZIP31*	Pav_sc0005841.1_g030.1.mk	S	639	212	24,188.51	11.1	Nucleus
*PavbZIP32*	Pav_sc0001341.1_g060.1.mk	B	510	169	18,978.08	10.31	Nucleus
*PavbZIP33*	Pav_sc0000396.1_g1160.1.mk	C	1344	447	48,357.08	6.51	Nucleus
*PavbZIP34*	Pav_sc0000110.1_g150.1.mk	D	1359	452	49,961.85	8.44	Nucleus
*PavbZIP35*	Pav_sc0008321.1_g030.1.mk	D	1635	544	61,074.37	7.19	Nucleus
*PavbZIP36*	Pav_sc0000363.1_g920.1.mk	A	2001	666	73,099.78	8.78	Nucleus
*PavbZIP37*	Pav_sc0001175.1_g270.1.mk	G	945	314	32,954.1	6.06	Nucleus
*PavbZIP38*	Pav_sc0000099.1_g260.1.mk	H	276	91	10,271.53	4.25	Nucleus
*PavbZIP39*	Pav_sc0000044.1_g090.1.mk	A	663	220	24,133.76	9.22	Nucleus
*PavbZIP40*	Pav_sc0000863.1_g670.1.mk	A	1791	596	65,091.38	6.8	Nucleus
*PavbZIP41*	Pav_sc0001518.1_g090.1.mk	S	225	74	7872.53	3.73	Nucleus
*PavbZIP42*	Pav_sc0002234.1_g130.1.mk	A	1326	441	47,583.34	10.23	Nucleus
*PavbZIP43*	Pav_sc0000407.1_g140.1.mk	E	576	191	22,444.78	10.16	Nucleus
*PavbZIP44*	Pav_sc0000983.1_g310.1.mk	C	2136	711	77,608.53	9.48	Nucleus
*PavbZIP45*	Pav_sc0000130.1_g590.1.mk	S	429	142	16,109.17	5.32	Nucleus
*PavbZIP46*	Pav_sc0000008.1_g430.1.mk	G	1512	503	54,115.73	8.97	Nucleus
*PavbZIP47*	Pav_sc0000322.1_g160.1.mk	I	1041	346	37,843.8	7.51	Nucleus
*PavbZIP48*	Pav_sc0000627.1_g120.1.mk	M	906	301	33,743.99	9.68	Nucleus
*PavbZIP49*	Pav_sc0003905.1_g010.1.mk	B	2217	739	79,096.1	6.43	Nucleus
*PavbZIP50*	Pav_sc0005505.1_g010.1.mk	A	435	144	16,328.36	10.65	Nucleus
*PavbZIP51*	Pav_sc0010299.1_g010.1.mk	S	639	212	24,188.51	11.1	Nucleus
*PavbZIP52*	Pav_sc0012575.1_g010.1.mk	I	1281	426	46,510.51	6.46	Nucleus
*PavbZIP53*	Pav_co4033649.1_g010.1.mk	A	918	305	34,317.71	8.45	Nucleus
*PavbZIP54*	Pav_co4037981.1_g010.1.br	S	609	202	23,331.75	6.68	Nucleus

## Data Availability

Data is contained within the article and Appendix A.

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
