# Peer review of "bZIP Transcription Factor PavbZIP6 Regulates Anthocyanin Accumulation by Increasing Abscisic Acid in Sweet Cherry"

_ijms, 2024, doi:10.3390/ijms251810207_

Round 1

Reviewer 1 Report

Comments and Suggestions for Authors

Presented manuscript describes the results showing a novel mechanism by which 25 PavbZIP6 mediates anthocyanin biosynthesis in response to ABA and contributes to understanding of the mechanism of bZIP gene in the regulation of anthocyanin biosynthesis in sweet cherry. Experiment is good-planed and the main text is well-organized. Methodology is described in detail with adequate references (except for the statistics that should be developed). Discussion is short, but on topic, however, with the cited literature not very up-to-date (could be up-dated). A big result from the anti-plagiarism report, but after taking a closer look, it mainly concerns standard phrases.

Technical problem: Figure 1 it is blurry and impossible to read.

Author Response

Comments 1: Discussion is short, but on topic, however, with the cited literature not very up-to-date (could be up-dated). 

Response 1: Thank you for pointing this out. We have updated the latest research progress and have been highlighted in red in lines 290-297, 312-313.

Comments 2: Figure 1 it is blurry and impossible to read.

Response 2: I am sorry for this! Thank you for pointing this out. We have redrawn Figure 1, and the new Figure 1 is on line 125.

Reviewer 2 Report

Comments and Suggestions for Authors

The statement according to which abscisic acid plays a vital role in the growth and regulation of different processes in plants, needs to be supported by additional examples in the introductory section. Current reference to ABA in the introduction is not sufficient. Moreover, please indicate different anthocyanins that can be present in cherries and include the description of their compounds’ structure.

Section 4.3. Determination of Anthocyanin Content. Please specify the amount of methanolic solution as well as hydrochloric acid used during the extraction process. Moreover, please indicate the amount of extract used in absorbance measurements.

All used acronyms such as EGY48, pCAMBIA1302, PavbZIP6, CaMV35S, PEG/LiAc need to be explained

Section Self-Activation Activity Analysis. Explain why the Salomon DNA was used in this experiment.

Figure 1. is unreadable. The same problem is with the all Figures – the font is illegible because it is too small.

Author Response

Comments 1: The statement according to which abscisic acid plays a vital role in the growth and regulation of different processes in plants, needs to be supported by additional examples in the introductory section. Current reference to ABA in the introduction is not sufficient.

Response 1: Thank you for your valuable suggestions on the manuscript.  We have added updates on the research progress related to ABA and have been highlighted in red in lines 87-88.

Comments 2: Please indicate different anthocyanins that can be present in cherries and include the description of their compounds’ structure.

Response 2:  Thank you for pointing this out. We have added the relevant content and have been highlighted in red in lines 49-52.

Comments 3: Section 4.3. Determination of Anthocyanin Content. Please specify the amount of methanolic solution as well as hydrochloric acid used during the extraction process. Moreover, please indicate the amount of extract used in absorbance measurements.

Response 3:  I am sorry for this! Thank you for pointing this out. We have corrected this part according to the requirements and have been highlighted in red in lines 362-369.

Comments 4: All used acronyms such as EGY48, pCAMBIA1302, PavbZIP6, CaMV35S, PEG/LiAc need to be explained.

Response 4:  I am sorry for this! Thank you for pointing this out. We have corrected this part according to the requirements and have been highlighted in red in lines 19,.380, 381, 385, 387.

Comments 5: Section Self-Activation Activity Analysis. Explain why the Salomon DNA was used in this experiment.

Response 5: Thank you for pointing this out. We have modified the salmon DNA into Carrier DNA and have been highlighted in red in line 385. Carrier DNA is a short, linear single-stranded DNA that can encapsulate plasmid DNA. It plays a role in facilitating the uptake of foreign plasmid DNA into yeast cells and additionally protects the plasmid from degradation by nucleases. 

Comments 6: Figure 1. is unreadable. The same problem is with the all Figures – the font is illegible because it is too small.

Response 6: I am sorry for this! Thank you for pointing this out. We have redrawn Figure 1, and the new Figure 1 is on line 125.